# Comparative Assessment of the Basic Chemical Composition and Antioxidant Activity of *Stevia rebaudiana* Bertoni Dried Leaves, Grown in Poland, Paraguay and Brazil—Preliminary Results

**Teresa Leszczyńska** [1,*], **Barbara Piekło** [1], **Aneta Kopeć** [1] and **Benno F. Zimmermann** [2]

1   Department of Human Nutrition, Faculty of Food Technology, University of Agriculture in Krakow, Balicka 122, 30-149 Kraków, Poland; barbara.bugaj@gmail.com (B.P.); aneta.kopec@urk.edu.pl (A.K.)
2   Institute of Nutritional and Food Sciences, Food Sciences, University of Bonn, Meckenheimer Allee 116a, 53115 Bonn, Germany; benno.zimmermann@uni-bonn.de
*   Correspondence: teresa.leszczynska@urk.edu.pl; Tel.: +48-12662-4814

**Abstract:** This study compares the content of basic nutrients (proteins, fats, digestible carbohydrates, dietary fiber and ash), steviol glycosides, selected antioxidants (vitamin C, total polyphenols) and antioxidant activity in dried leaves of *Stevia rebaudiana* Bertoni cultivated in Poland, Paraguay and Brazil and available in the direct sale. The basic chemical composition was determined by standard AOAC (Association of Official Analytical Chemists) methods. Content of steviol glycosides was determined by the UHPLC-UV chromatographic method. Total polyphenols content was expressed as gallic acid equivalent (GAE) and catechins equivalent (CE). Antioxidant activity was measured as ABTS$^{\bullet+}$ free radical scavenging activity. Dried leaves of *S. rebaudiana* grown in Poland had significantly higher contents of dietary fiber, and lower protein and ash content, compared to those derived from Paraguay and Brazil. The former had, however, considerably higher contents of total steviol glycosides, stevioside and rebaudioside D, compared to the remaining two plants. In the Paraguay-derived dried leaves, the content of rebaudioside A, C, E and rubusoside was found to be significantly lower. Dried leaves of *S. rebaudiana* Bertoni, cultivated in Poland, contained substantially more vitamin C and a similar content of total polyphenols, compared to those from Brazil and Paraguay. The examined material from Brazil and Paraguay plantations showed similar antioxidant activity, while that obtained from Polish cultivation was characterized by a significantly lower value of this parameter.

**Keywords:** *Stevia rebaudiana* Bertoni; steviol glycosides; antioxidant activity

## 1. Introduction

The increase in the incidence of chronic noncommunicable diseases, including atherosclerosis, high blood pressure and type 2 diabetes, which has been observed for many years, is mainly associated with obesity. The main cause of these diseases is an incorrect lifestyle, including an inappropriate diet. Numerous studies confirm that sugar, as a food ingredient, plays a role in the development of obesity and diabetes, and cardiovascular diseases [1–4]. Therefore, a search is still going on for new substances with high sweetening power and low calorie content, which may be an alternative to sugar. Intense sweeteners allow for the sucrose elimination from food products. Most are synthetic compounds.

For some time, researchers have been paying much attention to the *Stevia rebaudiana* Bertoni. The plant belongs to the *Asteraceae* family and is native to Brazil and Paraguay. For centuries, "sweet leaves" were used by the indigenous tribes of Paraguay and Brazil to counter the bitter taste of various plant-based medicines and drinks. The Guarani Indians used them to sweeten and stimulate digestion, and applied as compresses for wounds and other skin ailments [5].

This plant attracted great interest of nutritionists due to compounds of strong sweetening properties present in its leaves as diterpene glycosides, rebaudioside, which is 250–300 times sweeter than sucrose. Studies have also shown that *S. rebaudiana* is a good source of total carbohydrates (including dietary fiber), minerals and contains a number of secondary metabolites, such as pigments or phenolic compounds. The latter, with a role other than nutritional, are not considered essential for the body functioning. However, they may prevent diseases. Their systematic intake may have a beneficial effect on health protection and prevent or even treat diseases [6–8].

Essential oils, sesquiterpenes (δ-caryophyllene, δ-farnesene, humulene, candinene, nerolidol) and monoterpenes (linalool, terpinen-4-ol, terpineol) are important phytochemical compounds of this plant. The leaves also have a significant amount of tannins and alkaloids, moderate contents of cardiac glycosides and saponins, and small amounts of anthraquinones. Moreover, the dry-leaf extract contains water-soluble xanthophylls and chlorophylls, and hydroxycinnamic acids (caffeic and chlorogenic acid) [9–14].

Studies indicate that steviol glycosides derived from *S. rebaudiana* leaves have, i.a., hypoglycemic [15–21], hypotensive [16,18,22–26], anti-inflammatory and immunomodulatory properties [16,21,27–29].

Steviol glycosides are widely used in many countries around the world, including Japan, Brazil, Australia, Switzerland, United States, Mexico and the European Union. In 2008, the United States Food and Drug Administration (US FDA) qualified steviol glycoside preparations as Generally Recognized as Safe (GRAS). In the same year, the Joint FAO/WHO Expert Committee on Food Additives (JECFA) established the Acceptable Daily Intake (ADI) of steviol glycosides at 0–4 mg/kg body weight/day [30,31].

In 2008, the European Union recognized *S. rebaudiana* as the novel food, but now as a regular food product [32,33]. In Germany in 2011, *S. rebaudiana* was established as a traditional food [34,35]. In order to ensure consumer safety, in 2011, the regulation concerning the using steviol glycosides (E 960) in food products was published. Steviol glycosides can be used as sweeteners in beverages, sweets, sweeteners, chewing gums, fruit and vegetable preserves, bakery products and fish preserves. E 960 sweeteners are not authorized as a food additive for infants, toddlers and children (under 3 years of age) [32]. In 2016, the EU gave detailed information and conditions for the using the name of steviol glycosides [32,36].

From the best knowledge of authors in the literature, there are few data concerning the comparison of chemical composition and steviol glycosides content in *Stevia* leaves cultivated in various countries. What is important, previously the composition of commercial *Stevia* products derived from different countries was not simultaneously compared in the same study.

Therefore, the aim of this study was to compare the content of basic chemical composition (fats, proteins, digestible carbohydrates, dietary fiber and ash), steviol glycosides, selected antioxidants (vitamin C, total polyphenols) and the antioxidant activity of dried leaves of *S. rebaudiana* from Polish, Paraguayan and Brazilian cultivations.

## 2. Materials and Methods

The experimental material consisted of dried *Stevia* leaves from domestic and foreign cultivation, i.e., Polish, Paraguayan and Brazilian. Polish dried leaves were bought directly from the producer via the Internet, but dried Paraguayan and Brazilian *Stevia* were purchased through an online store (Figure 1).

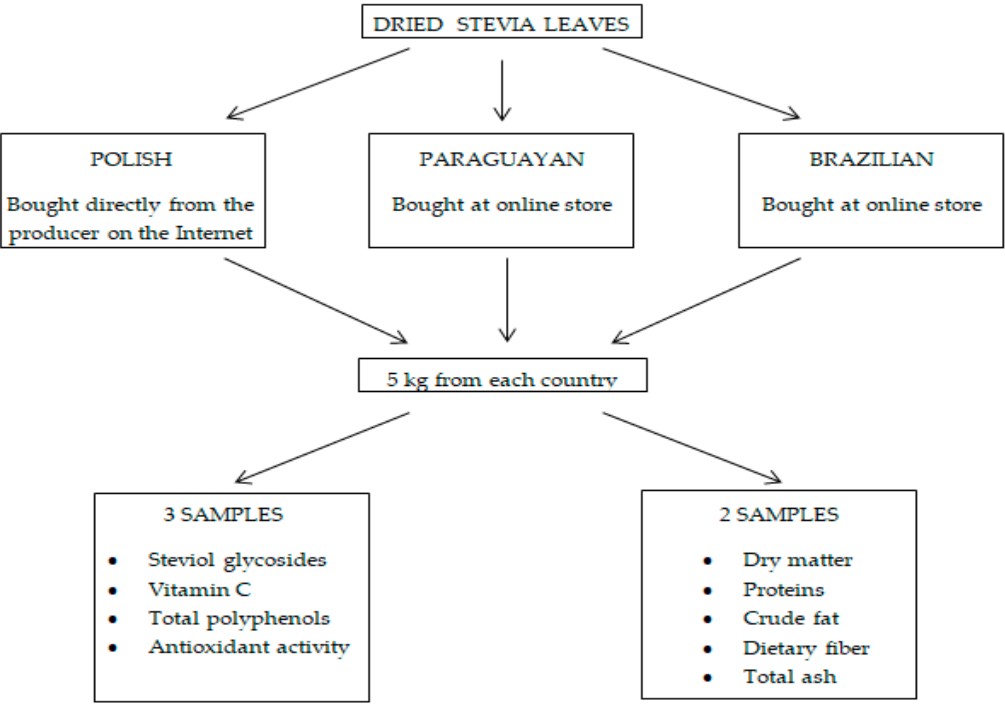

**Figure 1.** Scheme of obtaining research samples.

## 2.1. Proximate Composition

The basic chemical composition of the dried *Stevia* leaves was determined by standard AOAC methods [37]. The leaves were examined for dry matter: protein content by the Kjeldahl method using the FOSS Digester and Autodistillation Unit, KjeltecTM 8200 (Tecator Foss, Hillerød, Sweden); crude fat content by means of the Soxtec Avanti's 2050 Auto Extraction Foss Unit (Tecator, Hillerød, Sweden); and dietary fiber by enzymatic-gravimetric method. Crude ash content was determined at 525 °C (until constant weight).

### 2.1.1. Dry Matter

Dry matter content was determined by the weight loss after water removal from the product drying at 105 °C to a constant mass in a laboratory oven, in accordance with the AOAC 925.10 Official Method.

### 2.1.2. Total Protein

Total protein content was determined by the Kjeldahl's method, using the FOSS Digester and Autodistillation Unit, KjeltecTM 8200 (Tecator Foss, Hillerød, Sweden), in accordance with the AOAC 950.36 Official Method.

Approximately 0.5 g of the sample was weighed with 0.001 g accuracy and transferred, without losses, to a Kjeldahl digestion flask. Then, potassium sulfate (2.2 g) and copper sulfate (0.25 g) were added and poured together with 7 mL of concentrated sulfuric acid. This was placed in a heat block for about 1.5 h, until the solution reached a bright greenish color. After mineralization, the flask was cooled and placed in a nitrogen determination unit (2200 Kieltec Auto Distillation, Foss Tecator Chromo Serv). The resulting solution was titrated with 0.1 M HCl solution.

The percentage of total protein was calculated from the formula $X = (a \times 0.1 \times 14.007 \times 100/m) \times 6.25$, where X—total protein content (%), a—volume of acid used for sample titration (mL), m—sample weight (mg) and 6.25—conversion factor.

### 2.1.3. Crude Fat

Crude fat content was determined by the Soxhlet method, using the Soxtec Avanti's 2050 Auto Extraction Foss Unit (Tecator, Hillerød, Sweden), according to the AOAC 935.38 Official Method. The extraction cup was dried at 105 °C for 1 h in a laboratory dryer, cooled in a desiccator and weighed on an analytical balance. About 3 g of dried *Stevia rebaudiana* leaves were weighed into the extraction thimble with 0.0001 g accuracy. The thimble was covered with degreased cotton wool and inserted into the extraction unit (Universal Soxtec extractor 07-053). Then, about 80 mL of petroleum ether was added to the previously dried and weighed extraction cup and the extraction process started. At the end of the process, the cups were dried in a laboratory dryer (105 °C; 1 h) and reweighed.

The percentage of fat content (T) was calculated from the formula: $T = [(b - a) \, 100]/c$, where a—weight of the empty cup (g), b—weight of the cup with fat (g) and c—sample weight (g).

### 2.1.4. Total Ash

Total ash was determined according to the AOAC 930.05 Official Method. The porcelain crucible was placed in a muffle furnace (525 °C, 3 h). Then, it was cooled in a desiccator and weighed with 0.0001 g accuracy. About a 2 g sample of the examined material was weighed into a pre-weighted crucible, which was then placed over the gas burner in the laboratory fume cupboard, in order to pre-ash the sample. Afterwards, the crucible with the pre-incinerated sample was inserted into a muffle furnace (525 °C) and burned to constant weight (3 days). After cooling in the desiccator, the whole sample was reweighed.

The percentage of crude ash was calculated from the formula: $X = (b/a) \times 100$, were: X—crude ash content (%), a—sample weight (g) and b—ash weight (g).

### 2.1.5. Dietary Fiber

The content of dietary fiber was determined by the enzymatic–gravimetric method according to the AOAC 985.29 Official Method.

About 0.5 g of dried *Stevia rebaudiana* leaf sample was weighed into 200 mL conical flasks. Phosphate buffer (pH = $6.0 \pm 0.2$, 50 mL) was then added to each flask. After thoroughly mixing and checking pH (5–7), thermostable $\alpha$-amylase (50 μL) was added and the whole sample was mixed. The flasks were covered with aluminum foil, placed in a boiling water bath and incubated for 30 min. Afterward, the samples were removed from the water bath, cooled and adjusted to pH 7.3–7.7 by using about 10–15 mL of NaOH. Next, protease (100 μL) was added to each flask. The flasks were covered with aluminum foil again and incubated in a water bath at 60–62 °C for 30 min. Then, the samples were cooled to room temperature and adjusted to pH 4.3–4.7 with 0.325 N HCl (about 10 mL) in order to add amyloglucosidase (200 μL), which was followed by a 20 min incubation at 60 °C in a water bath. Then, 280 mL of 95% ethyl alcohol (60 °C) was added and samples were left for 1 h. The whole sample was filtered through a pre-weighted filter with the recorded weight, containing about 1 g of celite. The sediment was washed three times with 78% ethyl alcohol (20 mL) and twice with 95% ethyl alcohol (10 mL). The filters with sediment were placed overnight at 105 °C in the dryer. The next day, after cooling, they were weighed. The analysis was performed in duplicate. The residue from one filter was used for protein determination by the AOAC 950.36 method, while that from the other filter was used for determining ash content.

The content of dietary fiber per 100 g of the examined sample was calculated according to the following formula: dietary fiber content (%) = $\{[R \times (1 - P) - (A - C)]/W - B\} \times 100$, where R—weight of residue in the sample after drying (g), P—protein fraction, expressed as a decimal, A—filter weight + celite + ash (g), C—filter weight + celite + ash (g), B—weight of the blank (g) and W—sample weight (g).

### 2.1.6. Total and Digestible Carbohydrates

Total carbohydrate content was calculated by subtracting the sum of total fat, protein, moisture and ash from the wet matter, which was taken as 100%. Digestible carbohydrates were calculated by subtracting dietary fiber from total carbohydrate [38].

### 2.2. Steviol Glycosides

For chromatographic analysis of the single steviol glycosides, the *Stevia* leaves were ground in a domestic blender and 0.5 or 1 g of the ground leaves were extracted by adding 10 mL of acetonitrile + water (80 + 20, v + v) for 10 min in an ultrasonic bath. The extract was filtered through 0.2 μm Chromafil RC-20/15 MS filters (Macherey-Nagel, Düren, Germany) before HPLC analysis. The HPLC method is described in detail elsewhere [39]. The single steviol glycosides were quantified by external standards of stevioside and rebaudioside A by UV detection at 200 nm and the respective conversion factors for the other steviol glycosides [40]. The peaks were identified by retention time and by mass spectrometry in SIM mode by *m/z* of the $[M - H]^-$ ions. A typical chromatogram is shown in Figure 2.

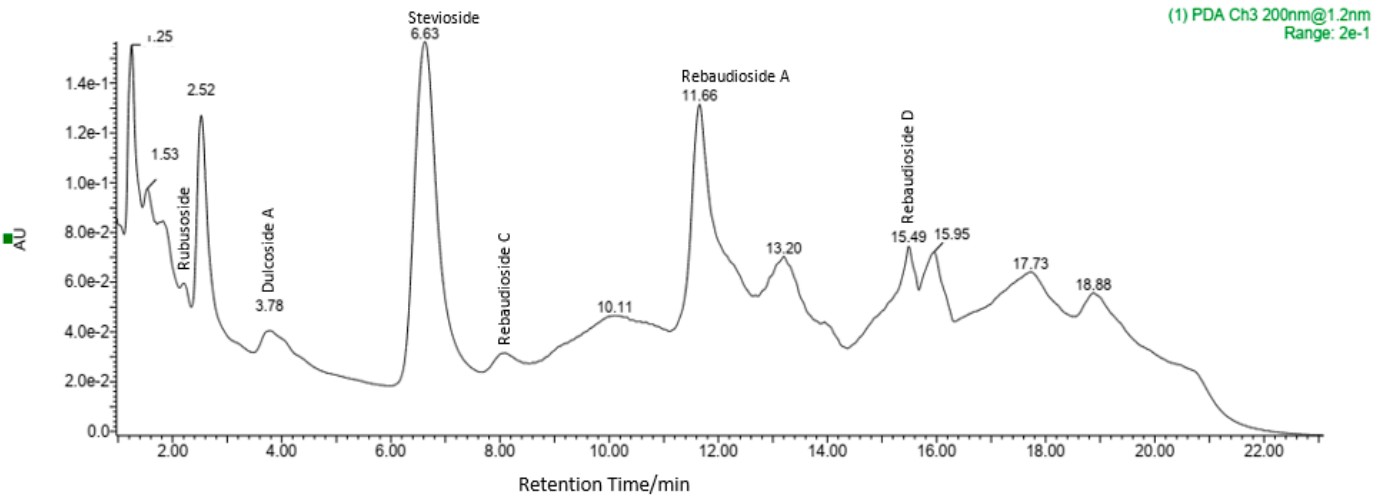

**Figure 2.** Typical chromatogram of a *Stevia* leaf extract recorded at a wavelength of 200 nm.

### 2.3. Vitamin C

Vitamin C content was measured by Tillmans method with Pijanowski modification. Based on this method, vitamin C is calculated as a sum of L-ascorbic and L-dehydroascorbic acids [41]. The principle of this method is the reduction of dehydroascorbic acid to ascorbic acid by sodium sulfide solution, followed by precipitation of excess sulfide with mercury chloride solution, and determining the sum of ascorbic acid in the material by means of titration with 2,6-dichlorophenolindophenol.

### 2.4. Preparation of Methanol Extracts

The ground material (5 g) was poured with 80 mL of 70% methanol (POCH, Gliwice, Poland) and shaken for 2 h in an Elpan water bath with a shaker (type 357, Lubawa, Poland). Afterward, the mixture was filtered and stored frozen at −20 °C until analysis [42].

### 2.4.1. Phenolic Compounds

In brief, from the previously prepared methanol extracts, dilutions (1:20) were prepared. In the test tube, 5 mL of the diluted extract, 0.5 mL of Folin–Ciocalteau reagent (POCH, Gliwice, Poland) (diluted with distilled water in the ratio 1:1) and 0.25 mL of 25% sodium carbonate solution (POCH, Gliwice, Poland) were mixed and left for 20 min in darkness. Then, the sample was centrifuged (1250 rpm for 10 min) and the absorbance at 760 nm measured using a spectrophotometer (RAYLEIGH, UV-1800, Beijing Beifen-Ruili

Analytical Instrument Co., Ltd., Beijing, China), compared to 70% methanol. Results were expressed as gallic acid equivalents (GAE) and catechin equivalents (CE) [43].

### 2.4.2. Antioxidant Activity

The overall antioxidant potential was estimated by the ABTS$^{\bullet+}$ free radical scavenging assay [44]. Appropriate volumes of methanol extract were poured into test tubes and filled to 1 mL with methanol. Then, 2 mL of ABTS$^{\bullet+}$ free radical solution was added and the mixture incubated at 30 °C for 6 min. The absorbance at 734 nm was measured against methanol with a UV-1800 spectrophotometer (RAYLEIGH, UV-1800, Beijing Beifen-Ruili Analytical Instrument Co., Ltd., Beijing, China). Antioxidant activity was expressed as TEAC (Trolox equivalent antioxidant capacity).

Analyses of the basic chemical composition were made in duplicate, but steviol glycosides, vitamin C, phenolic compounds and antioxidant activity were done in triplicate (Figure 1).

### 2.5. Statistical Analysis

One-way analysis of variance (ANOVA) was applied to find the significance of differences between the content of analyzed components and antioxidant capacity of the dried *Stevia* leaves, depending on the country of origin. The significance of differences was assessed using Duncan's multiple range test at the critical level of significance $p \leq 0.05$. For the mean values obtained, standard deviation (SD) was also calculated.

## 3. Results and Discussion

### 3.1. Proximate Composition

#### 3.1.1. Dry Matter

This study revealed that dried leaves of *S. rebaudiana* Bertoni, from foreign and domestic crops, had a similar (~99.5 g/100 g) dry matter content ($p > 0.05$) (Table 1).

**Table 1.** Proximate composition of *Stevia rebaudiana* Bertoni dried leaves depending on the country of origin.

| Ingredient | *Stevia rebaudiana* | | |
|---|---|---|---|
| | **POLISH** | **PARAGUAYAN** | **BRAZILIAN** |
| Dry matter (g/100 g) | 99.6 ± 0.06 [a] | 99.4 ± 0.27 [a] | 99.3 ± 0.05 [a] |
| Protein (g/100 g DM) | 11.2 ± 0.97 [a] | 17.1 ± 1.85 [b] | 15.5 ± 0.51 [b] |
| Fat (g/100 g DM) | 3.33 ± 0.10 [b] | 1.73 ± 0.01 [a] | 4.90 ± 0.32 [c] |
| Digestible carbohydrates (g/100 g DM) | 61.7 ± 0.67 [b] | 60.4 ± 1.52 [b] | 55.0 ± 0.39 [a] |
| Dietary fiber (g/100 g DM) | 15.7 ± 0.18 [c] | 12.3 ± 0.3 [a] | 14.1 ± 0.22 [b] |
| Total ash (g/100 g DM) | 7.65 ± 0.40 [a] | 7.85 ± 0.03 [b] | 9.78 ± 0.05 [c] |

Letters [a], [b] and [c] indicate statistically significant differences ($p \leq 0.05$); data are shown as the mean ± SD (standard deviation).

The values reported in the literature were lower: 95.4 [45], 94.6 [6], 93.0 [7,46] and 92.3 g/100 g [47]. As all analyses of dry matter content were conducted in the dried material, the differences observed may result, for example, from different growth conditions and the various techniques and parameters applied for drying a fresh plant [5].

#### 3.1.2. Proteins

In comparison with two remaining *Stevia* sources, the dried leaves of *S. rebaudiana* originating from Polish cultivation contained significantly ($p \leq 0.05$) less protein (11.2 g/100 g DM). However, in the material derived from Paraguay and Brazilian cultivation, the content of this component did not differ significantly and amounted to 17.1 and 15.5 g/100 g DM, respectively (Table 1).

Protein contents in dried leaves of S. *rebaudiana* reported by other authors were 11.4, 11.2, 10.0, 11.2, and 9.8 g/100 g DM [6,7,45,46,48]. In view of this, only the value determined

in the dried *Stevia* leaves from Poland was within the range of the values reported by these authors. In turn, the highest protein content, amounting to 20.4 g/100 g DM, was found in this plant grown in India [49]. Taking into account the upper threshold cited above, the dry matter contents determined in dried *Stevia* leaves from Paraguay and Brazil were within the broad limits identified by the aforementioned authors.

According to Kaushik et al. [47], protein content depended on the harvest date and was 12.0 g/100 g DM in the plants harvested in summer season and 12.9 g/100 g DM in those from the autumn period. Our study revealed that the results obtained for the dried *Stevia* leaves from foreign crops were higher than those cited above. These differences may depend primarily on growing conditions (climate, soil composition, fertilization), storage conditions and the drying method used [5].

Additionally, in some scientific papers the amino acids in *Stevia* leaves were determined. Muhammed et al. [50] isolated nine amino acids from *Stevia* leaves, including glutamic acid, aspartic acid, lysine, serine, isoleucine, alanine, proline, tyrosine and methionine. In turn, Abou-Arab et al. [6] supplemented these data, in which seventeen amino acids, both exo- and endogenous, were identified.

### 3.1.3. Crude Fat

Crude fat content in *S. rebaudiana* Bertoni dried leaves was 3.33 g/100 g DM (Polish cultivation), 1.73 g/100 g DM (Paraguayan cultivation) and 4.9 g/100 g DM (Brazilian cultivation). The samples examined differed significantly ($p \leq 0.05$) in the level of this constituent (Table 1).

In the available literature, the highest fat content in the *Stevia* dry matter (5.6 g/100 g DM) was reported by Serio [48], while the lowest content (1.9 g/100 g DM) was found by Goyal et al. [45]. The results obtained by the remaining authors ranged between 2.5 and 4.34 g/100 g DM [6,7,46,47,49].

In view of these data, the dried *Stevia* leaves examined from Polish and Brazilian cultivation fell within the range of the results cited above.

According to Tadhani and Subhash [49] and Atteh et al. [51], dried leaves of *S. rebaudiana* contain a small amount of fat (about 4%). The authors identified six fatty acids, among which palmitic acid (27.5 and 29.5% respectively) was predominant, followed by linolenic acid (21.6 and 32.6%) and linoleic acid (12.4 and 16.8%). The remaining were palmitoleic, stearic and oleic acids.

### 3.1.4. Digestible Carbohydrates

The level of digestible carbohydrates in the dried leaves of *S. rebaudiana* from Polish cultivation was found to be similar (61.7 g/100 g DM; $p > 0.05$) as compared to those from Paraguayan (60.4 g/100 g DM) and significantly higher (55.0 g/100 g DM; $p \leq 0.05$) to those from Brazilian crops (Table 1).

According to the available literature, carbohydrate content in the dried *Stevia* leaves fluctuates in broad range, from 35.5 to 61.9 g/100 g DM. The values reported are 35.5, 46.1, 52, 53 and 61.9 g/100 g DM, by Tadhani and Subhash [49], Shivanna et al. [52], Mishra et al. [7], and Savita et al. [46], Serio [48] and Abou-Arab et al. [6], respectively. These are similar or smaller than the results obtained in this study for three plants, differing in origin: Polish, Brazilian and Paraguayan.

### 3.1.5. Dietary Fiber

The fiber contents in the dried *Stevia* leaves were 15.7 g/100 g DM (Polish), 12.3 g/100 g DM (Paraguayan), and 14.1 g/100 g DM (Brazilian). Statistical analysis revealed significant differences ($p \leq 0.05$) in the samples examined (Table 1).

The results obtained were lower compared to those cited in the literature. According to Mishra et al. [7] and Savita et al. [46], these values are 18.0 and 18.5 g/100 g DM, respectively. The fiber content determined in dried *Stevia* leaves from Polish cultivation

is consistent with the findings of Abou-Arab et al. [6], Goyal et al. [45], Serio [48] and Shivanna et al. [52], who noted values ranging from 15.0 to 15.9 g/100 g DM.

The benefits associated with the dietary fiber are due to its pre-biotic effect that promotes proliferation of beneficial intestinal microflora. *Stevia* roots and leaves have 4.6% fructooligosaccharides and polysaccharides, which are involved in lipid metabolism and control the level of blood sugar [53].

### 3.1.6. Total Ash

Ash content determined in the dried leaves of *S. rebaudiana* was 7.65 g/100 g DM (Polish), 7.85 g/100 g DM (Paraguayan), and 9.78 g/100 g DM (Brazilian). There were significant differences ($p \leq 0.05$) between the samples examined (Table 1).

According to the available literature, ash content in this plant differs strongly. The reported values are 7.4, 11.0, 10.5 and 8.4 g/100 g DM [6,7,46,47], respectively. The highest ash content (13.1 g/100 g DM) was determined by Tadhani and Subhash [49] in the dried leaves of *S. rebaudiana* Bertoni grown on an Indian plantation; the lowest (6.3 g/100 g DM of leaves) was found by Goyal et al. [45].

Thus, the ash content determined in the dried *Stevia* leaves derived from Polish, Paraguayan and Brazilian cultivation are within the range reported in the cited literature.

According to some authors, ash comprises about 11% of DM, and its main mineral constituents (in units of mg/100 g DM) are potassium 839–2510, calcium 464.4–1550, magnesium 349–500, sodium 14.93–190.0, and phosphorus 11.4–350.0 [11,49]. On the other hand, as noted by other authors, trace elements such as copper, cobalt, iron, manganese, zinc, selenium and molybdenum are present in minute amounts [11,49]. Based on the literature data [11], *S. rebaudiana* contains a high level of oxalic acid (2295 mg/100 g DM), which reduces human bioavailability of calcium, iron and other minerals.

### 3.2. Steviol Glycosides

In this work, nine sweet steviol glycosides were identified, the content of which in dried leaves may depend on growth and cultivation conditions. The sum of steviol glycosides, and stevioside and rebaudioside D content were significantly higher ($p \leq 0.05$) in the dried leaves of *Stevia* from Polish cultivation, compared to the two remaining crops. In turn, in the dry-leaf material of *S. rebaudiana* Bertoni from Paraguay, the amounts of rebaudioside C, rebaudioside A, rebaudioside E and rubusoside were significantly lower ($p \leq 0.05$), when compared to those from the other two crops (Table 2).

**Table 2.** Concentration of steviol glycosides (mg/100 g DM) in dried leaves of *Stevia rebaudiana* Bertoni, depending on the country of origin.

| Ingredient | *Stevia rebaudiana* | | |
|---|---|---|---|
| | **POLISH** | **PARAGUAYAN** | **BRAZILIAN** |
| Rubusoside | 1.13 ± 0.11 [a] | 0.53 ± 0.06 [b] | 1.18 ± 0.01 [a] |
| Steviolbioside | n.q. | n.q. | n.q. |
| Dulcoside A | 0.31 ± 0.04 [a] | 0.22 ± 0.01 [b] | 0.25 ± 0.02 [ab] |
| Rebaudioside B | n.q. | n.q. | n.q. |
| Stevioside | 9.54 ± 0.43 [b] | 5.94 ± 0.48 [a] | 5.82 ± 0.13 [a] |
| Rebaudioside C | 0.28 ± 0.01 [b] | 0.22 ± 0.03 [a] | 0.29 ± 0.01 [b] |
| Rebaudioside F | n.q. | n.q. | n.q. |
| Rebaudioside A | 4.05 ± 0.03 [b] | 3.01 ± 0.51 [a] | 4.35 ± 0.03 [b] |
| Rebaudioside E | 2.18 ± 0.10 [a] | 0.64 ± 0.26 [b] | 1.13 ± 0.02 [a] |
| Rebaudioside D | 4.64 ± 0.26 [c] | 2.10 ± 0.12 [b] | 0.47 ± 0.03 [a] |
| ∑ Steviol Glycosides | 22.14 ± 0.10 [b] | 12.67 ± 0.29 [a] | 13.5 ± 0.10 [a] |

Letters [a, b] and [c] indicate statistically significant differences ($p \leq 0.05$); data are shown as the mean ± SD (standard deviation); n.q.—not quantifiable.

The content of steviol glycosides, determined in this work, was in the range described by Geuns [54], 4–20% DM. In turn, Gardana et al. [55] and Goyal et al. [45] determined stevioside content at the level of 5.8% and 9.1% DM, respectively. This means that the content of this component detected in the Paraguayan and Brazilian *Stevia* leaves was close to the first of these values, while the value found for the Polish crop was similar to the second value.

Other authors [40,56], similar to this paper, indicate that the total mass of steviol glycosides extracted from *S. rebaudiana* leaves was dominated by stevioside and rebaudioside A, comprising approximately 65 and 25%, respectively; the remaining were rebaudiosides B, C, D, E, F, dulcoside A and steviolbioside.

### 3.3. Antioxidants and antioxidant activity

#### 3.3.1. Vitamin C

In this study, the highest vitamin C content (29.9 mg/100 g DM) was found in dried *Stevia* leaves from Polish crops; the average content of this vitamin, 4.55 and 7.58 mg/100 g DM, was in plants from Paraguay and Brazilian plantations, respectively. There were significant differences between these values ($p \leq 0.05$) (Table 3).

**Table 3.** Concentration of selected antioxidants in dried leaves of *Stevia rebaudiana* Bertoni depending on the country of origin.

| Ingredient | *Stevia rebaudiana* | | |
| :--- | :---: | :---: | :---: |
| | **POLISH** | **PARAGUAYAN** | **BRAZILIAN** |
| Vitamin C (mg/100 g DM) | 29.9 ± 0.53 [c] | 4.55 ± 0.01 [a] | 7.58 ± 0.21 [b] |
| Total polyphenols (mg GAE */g DM) | 89.96 ± 1.60 [a] | 89.31 ± 1.65 [a] | 90.95 ± 6.93 [a] |
| Total polyphenols (mg CE **/g DM) | 99.71 ± 1.77 [a] | 98.98 ± 1.83 [a] | 100.8 ± 7.68 [a] |
| Antioxidant activity (μmol Trolox/g DM) | 152.4 ± 2.36 [a] | 163.7 ± 2.00 [b] | 160.3 ± 1.56 [b] |

Letters [a, b] and [c] indicate statistically significant differences ($p \leq 0.05$); data are shown as the mean ±SD (standard deviation); * gallic acid equivalent; ** CE expressed as catechins equivalent.

Vitamin C content reported by Shivanna et al. [52] in dried leaves of *S. rebaudiana* obtained from an Indian plantation was at the level of 10 mg/100 g DM. In turn, the content of vitamin C determined in the water extract of dried leaves of *S. rebaudiana*, grown in Korea, was 15.0 mg/100 g DM [57].

Vitamin C is a nutrient that is very sensitive to physicochemical conditions, and differences between our results and other authors could be connected mainly with the drying method and condition of storage of this plant, and with another factor noted below.

#### 3.3.2. Total Polyphenols

The dried leaves of *S. rebaudiana* had similar total phenolic contents: 89.96 mg GEA/g DM and 99.71 mg CE/g DM (Polish); 89.31 mg GEA/g DM and 98.98 mg CE/g DM (Paraguayan); and 90.95 mg GAE/g DM and 100.8 CE/g DM (Brazilian) (Table 3).

The results obtained are higher than those given in the literature. According to Tadhani et al. [14], who applied the same Folin-Ciocalteu method to determine the total content of polyphenols and flavonoids in methanol extracts of *S. rebaudiana*, the level of polyphenols and flavonoids was 25.2 and 21.7 mg/g DM, respectively. These values are very close to those given by Muanda et al. [58], who used both methanol and aqueous extracts. In turn, total polyphenol content (expressed as catechin equivalent per weight unit of the sample), reported by Kim et al. [57] in an aqueous extract, was 130.7 mg CE/g DM. The value determined by the same method by Shukla et al. [59], but in ethanol extract of *Stevia* leaves, was 61.50 mg/g DM. On the other hand, Shivanna et al. [52] used the liquid chromatography method (HPLC) to establish total polyphenol and flavonoid content, which were 91 and 23 mg/g DM, respectively.

Of the *Stevia*-derived polyphenols, the following should be mentioned: phenolic acids (coumaric, sinapic and cinnamic acids), chlorogenic acids, flavonols (quercetin and kaempferol) and flavones [8].

### 3.3.3. Antioxidant Activity

In this study, the ABTS$^{\bullet+}$ free radical scavenging activity was applied to measure antioxidant activity of methanolic extracts obtained from the dried *Stevia* leaves. The antioxidant activity determined was 152.4, 163.7, 160.3 µmol Trolox/g for plants from Polish, Paraguayan and Brazilian plantations, respectively. The dried leaves originated from Polish crops differed significantly from those derived from two other plantations ($p \leq 0.05$) (Table 3).

Differences between high antioxidant activity in Paraguayan and Brazilian *Stevia* and, at the same time, lower content of polyphenolic compounds and vitamin C, compared to Polish *Stevia*, can be explain by higher content of other antioxidants that were not measured, for example, essential oils (sesquiterpenes-δ-caryophyllene, δ-farnesene, humulene, candinene, nerolidol and monoterpenes—linalool, terpinen-4-ol, terpineol), chlorofil, xanfofils and others [9–14].

According to the literature, the antioxidant activity of the aqueous, methanolic and ethanolic extracts of *S. rebaudiana* leaves is examined by means of the FRAP (ferrum reducing antioxidant power) and the DPPH$^{\bullet}$ (1,1-diphenyl-2-picrylhydrazyl) radical scavenging assays. The scavenging of the DPPH$^{\bullet}$ free radical, reported in the *Stevia* leaf extracts are 68.7, 56.8 and 39.8% for water, methanol and ethanol solvents, respectively, at a 600 µg/mL sample dilution [59,60]. In turn, the degree of DDPH$^{\bullet}$ free radical scavenging measured by Shukla at al. [61] and Rao [62] in water, methanol and ethanol extracts from *S. rebaudiana* leaves were 64.3, 52.5, and 62.8%, respectively, at a sample dilution of 100 µg/mL. When analyzed by the FRAP method, the extracts showed similar antioxidant activity [14,59]. On the other hand, Muanda et al. [58], who used water-methanol extract, noted the highest degree (96.9%) of free radical scavenging.

The values of the antioxidant activity of *S. rebaudiana*, which the aforementioned authors determined by other methods, differ from those obtained in this study. This may result from both the advantages of the method used and its disadvantages. For example, the method with the ABTS$^{\bullet+}$ reagent is one of the most precise, because it allows for the measurement of the total antioxidant activity of all antioxidants, both lipophilic and hydrophilic. The value obtained, expressed by the rate of radical reduction (change in color), depends both on the antioxidant concentration in the sample and its power. On the other hand, the DPPH method allows for only the determination of hydrophilic antioxidants [63].

As shown in this study and in the publications cited, the composition of *S. rebaudiana* ranges widely. Cultivation environmental factors such as temperature, sunlight exposure, soil composition and pH, fertilization, planting density, growth phase and plant physiology strongly affect chemical composition. Additionally, drying method and storage conditions can modify bioactive compounds content [5,64].

### 4. Conclusions

*S. rebaudiana*, grown in Poland, had significantly higher content of dietary fiber, and lower protein and ash content in the dried leaves, compared to the plants from Paraguayan and Brazilian cultivation. Dried leaves of *S. rebaudiana* from Polish crops contained substantially more total steviol glycosides, stevioside and rebaudioside D, compared to the leaves of plants from the two other origins. The content of rebaudioside A, C, E and rubusoside was markedly lower in Paraguay-derived plants. *S. rebaudiana* planted in Poland had a considerably higher content of vitamin C and similar content of total polyphenols, compared to the plants cultivated in Brazil and Paraguay. Antioxidant activity determined in dried leaves of *S. rebaudiana* originating from foreign crops was found to be similar. The value of this parameter was significantly lower in the leaves from Polish crops. Results of

this research could be used in future animal or human studies to evaluate, for example, antioxidant and anti-inflammatory properties.

**Author Contributions:** Conceptualization, T.L. and B.P.; methodology, B.P., A.K. and B.F.Z.; formal analysis, B.P. and B.F.Z.; writing—original draft preparation, T.L., B.P. and A.K.; writing—review and editing, T.L., B.P., A.K. and B.F.Z.; supervision, T.L.; resources and funding acquisition, T.L. and A.K. All authors have read and agreed to the published version of the manuscript.

**Funding:** The study was financed by the Ministry of Science and Higher Education of the Republic of Poland.

**Institutional Review Board Statement:** Not applicable.

**Informed Consent Statement:** Not applicable.

**Data Availability Statement:** Not applicable.

**Conflicts of Interest:** The authors declare no conflict of interest.

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
