# Peer review of "Comparative Assessment of the Basic Chemical Composition and Antioxidant Activity of Stevia rebaudiana Bertoni Dried Leaves, Grown in Poland, Paraguay and Brazil—Preliminary Results"

_applsci, doi:10.3390/app11083634_

Round 1
Reviewer 1 Report
The authors compare the major and minor composition of Stevia from various origins. The methods are described in a fashion allowing reproducibility. The results are compared to a number of previous investigations of Stevia.
Some minor comments:
Line 15: is the content “estimated” or “determined”? I believe HPLC typically allows more than an estimation?
Line 35: sugar is not normally treated as a food additive, at least not in a legal sense
Line 59: spelling caffeic acid
Line 70: The status of Stevia as novel food in the EU should be updated. Currently, the leaves are not novel (check EU novel food catalogue).
Line 77: Check relevance, this is for mustard only?
Section 2.1: can more information about the samples be provided? Origin, processing, storage, etc.. How many samples per origin per analyses? Are the standard deviation in the table for replicates of one sample or for several different samples per origin?
Line 103: can a reference for “elsewhere” be provided? Is this 39 as well?
Line 255: check grammar. “really unstable” is unscientific
Line 259: add space before had
Tables 1-3: please make the layout of tables consistent
References: check style. Should DOI be added?
Line 373: do not underline title
Line 377/380: do not use 1995ab in numbered references
Author Response
Reviewer 1
Authors are grateful for the Reviewer comments. The answer to the comments are below.
Line 15: “is the content “estimated” or “determined”? I believe HPLC typically allows more than an estimation?”
Now line 15-16
Corrected
It was estimated, now it is determined
Line 35: “sugar is not normally treated as a food additive, at least not in a legal sense”
Corrected
It was additive, now it ingredient
Line 59: “spelling caffeic acid”
Now line 60
Corrected
It was coffeic, now is caffeic
Line 70: “The status of Stevia as novel food in the EU should be updated. Currently, the leaves are not novel (check EU novel food catalogue)”.
Now line 71
Corrected
It was novel, now is regular food products, and proper references has been added [33]
Line 77: “Check relevance, this is for mustard only?”
Now line 78
Corrected
One more references has been added [32]
Section 2.1: “can more information about the samples be provided? Origin, processing, storage, etc..”
We researched dried stevia leaves, as a ready commercial product, that were purchased from Poland direct trade points via the Internet - Polish dried stevia directly from the producer, but dried Paraguay and Brazilian stevia through an online store. Shops run commercial activities, therefore mentioning the company names would be inappropriate, because of free advertising for them.
Line 89-91. Some information has been added in manuscript body - Polish dried leaves were bought directly from the producer via the Internet, but dried Paraguay and Brazil stevia through an online store.
“How many samples per origin per analyses? Are the standard deviation in the table for replicates of one sample or for several different samples per origin?”
Line 133-134. Some information has been added in manuscript body - Analysis concerning the basic chemical composition were made in duplicate, but steviol glycosides, vitamin C, phenolic compounds and antioxidant activity in triplicate.
Line 103: can a reference for “elsewhere” be provided? Is this 39 as well?
Now line 107
Corrected
Line 255: check grammar. “really unstable” is unscientific
Now line 267
Corrected
It was “rally unstable”, now is “sensitive nutrient to physicochemical conditions”
Line 259: “add space before had”
Corrected
“Tables 1-3: please make the layout of tables consistent”
Corrected
“References: check style. Should DOI be added?”
Corrected
DOI has been added
Line 373: “do not underline title”
Now line 398
Corrected
Line 377/380: “do not use 1995ab in numbered references”
Now line 402-405
Corrected
Reviewer 2 Report
This manuscript compared the chemical composition and antioxidant activity of Stevia rebaudiana from different source of origins. By utilizing the standard analytical methods, authors listed the major differences of compounds concentration, and compared with reported data from other references. However, this paper contains many questionable flaws in the experimental designs and data processing. The reviewer would like the authors to address the questions listed below:
- The samples were purchased from the direct sales in three countries. There were no data indicating which area of the countries, the season, and the criteria of how the samples were selected. The reviewer wants the authors to justify how were the samples representative for the overall population of the different cultivation countries? If the samples were selected in large population and evenly distributed, it would be not convincible to claim the results in the nation-wide comparison.
- How is the batch-to-batch variation within the certain country? The consistency of samples within the same country should be good enough in order to make comparison among different countries.
- The data were shown in the mean and standard deviation. However, it was not acceptable to miss the information of the number of replicates. It is very basic knowledge in statistical analysis.
- The paper simply put the comparison results together. What were exhibited is the concentration or level of certain compounds in certain samples, but without any further analysis. What information or suggestions can readers get from this paper?
- In Figure 1, the peaks were not assigned. Is the wavelength 220nm shown as the legend or 200nm shown on the right top? In the paragraph that cited Figure 1, (Line 234), where was the comparison? Are the numbers 803.8 and 1127.8 molecular weight? If so, please provide the MS spectra of the other peaks.
- Line95-96: Please specify the sentences described “how the digestible carbohydrate content was calculated” in the reference 38.
- Line 103: add references for “elsewhere”.
- Please provide the cited sentences in those references that support the data in Line 140-141.
- Line 165-168, Did authors isolate free amino acids as well in this paper? If not, what is the purpose of introducing others’ work.
- Line 223-224, did authors test the trace elements? What methods were used to detect those trace elements reported as “present”?
- Line 225-226, was the oxalic acid detected from the ash content?
- Table 1, digestible carbohydrates: Paraguayan stevia (60.4 ±52) and Brazilian stevia (55.0 ± 0.39) are in the group a, but Polish stevia (61.7 ± 0.67) is in the group b? Why is Brazilian group more similar to Paraguayan than the Polish group?
To sum up, this manuscript failed to show an innovative and representative research. The significance of the study was not highlighted, so the reviewer could not get its purpose. Moreover, there are big flaws in the experimental design and the figures, as well as the reference citations. The reviewer considers this manuscript not suitable to be published.
Reviewer 3 Report
The novelty of the study is dubious. It is described in lines 78-80 as: „For the best knowledge of authors in the literature, there are not many data concerning the comparison of chemical composition and steviol glycosides (SGs) content in stevia leaves cultivated in various country.” There are many data concerning the chemical composition of stevia cultivated in different countries for different kind of studies. The authors aim to compare stevia cultivated in Poland, Paraguay and Brazil, however the origin of samples, i. e. the prove for its cultivation country, has been missing since authors do not give information on product manufacturers or sellers (see line 88, “available in direct sales”). Also, the cultivation conditions are unknown (probably greenhouse for Poland and open field for Paraguay and Brazil?). This information is not crucial for product characterization but could be helpful for result interpretation.
The Introduction is lacking a structure (i. e. there is a paragraph about essential oil of stevia, however, it is not an object in the current study, etc.).
In the Methods, vitamin C determination is included for some reason in „Proximate composition“ chapter and only reference for calculation, but not measurement method of vitamin C is given. Two different sample amounts are given in extraction procedure without reasonable explanation (line 100). There is no reference given for HPLC method. Reference 39 in line 105 is erroneous. From the description, it is not clear weather MS spectrometer is used only for identification or for both – identification and quantification of SGs. The chromatogram in Figure 1 does not give any information since the peaks of SGs and the integration baseline are not marked. Reference for Figure 1 (line 234) states differences in SGs concentrations however Figure 1 has nothing to do with this information. Centrifugation units in rpm (line 117) are useless unless rotor dimensions or centrifuge model is given. The given result units (GAE) are different in result chapter (GAE/g DM).
In the Results and Discussion, unrelated information is given in the last paragraphs of chapters 3.1.2-3, 3.1.6. Radical ABTS abbreviation is given in three different forms in lines 18, 122, 124. Also, two terms are used for the same parameter – radical scavenging and radical inhibition (lines 286,289). The terms „fat“ and „lipids“ are used as equivalents (lines 169, 170). SGs rebaudiosides B and F, steviolbioside were not detected by authors, however they are stated as „remaining“, i. e. present after dominant stevioside and rebaudioside A.
Author Response
Reviewer 3
Authors are grateful for the Reviewer comments. The answer to the comments are below.
- “The novelty of the study is dubious. It is described in lines 78-80 as: „For the best knowledge of authors in the literature, there are not many data concerning the comparison of chemical composition and steviol glycosides (SGs) content in stevia leaves cultivated in various country.” There are many data concerning the chemical composition of stevia cultivated in different countries for different kind of studies”.
We agree that “There are many data concerning the chemical composition of stevia cultivated in different countries for different kind of studies”, but there is no work which compare at the same time the chemical composition those kind of material from different countries.
- “The authors aim to compare stevia cultivated in Poland, Paraguay and Brazil, however the origin of samples, i. e. the prove for its cultivation country, has been missing since authors do not give information on product manufacturers or sellers (see line 88, “available in direct sales”). Also, the cultivation conditions are unknown (probably greenhouse for Poland and open field for Paraguay and Brazil?). This information is not crucial for product characterization but could be helpful for result interpretation”.
- We had researched dried stevia leaves, as a ready to use commercial product, that were purchased from Poland in direct trade points via the Internet. Polish dried stevia directly was bought from the producer, but dried Paraguay and Brazilian stevia through an online store. The seller were able give the country of origin of stevia but not the part of region were the stevia was agricultured. Shops run commercial activities, therefore mentioning the company names would be inappropriate, because of free advertising for them.
Line 89-91. Some information has been added in manuscript body - Polish dried leaves were bought directly from the producer via the Internet, but dried Paraguay and Brazil stevia through an online store.
- “The Introduction is lacking a structure (i. e. there is a paragraph about essential oil of stevia, however, it is not an object in the current study, etc.).”
Corrected
Line 54. Essential oils Lipid fraction of Stevia, being contains sesquiterpenes
- “In the Methods, vitamin C determination is included for some reason in „Proximate composition“ chapter and only reference for calculation, but not measurement method of vitamin C is given”.
Corrected
Line 97-99. Vitamin C content was measured by Tillmans method with Pijanowski modification. Based on this method vitamin C is shown calculated as a sum of ascorbic and dehydroascorbic acids [37 38].
- “Two different sample amounts are given in extraction procedure without reasonable explanation (line 100)”.
Two different amounts were weightned in order to match the calibration. If the peaks were too big in the extract from 1 g, the peaks had the right size, i.e., within the calibration range in the extract of the lower amount.
- “There is no reference given for HPLC method. Reference 39 in line 105 is erroneous”.
In reference 39 (now 40) at page 1576 the HPLC method is described in detail. But the reference to the article describing the conversion factors was missing. We’re sorry for the mistake and added the reference.
- “From the description, it is not clear weather MS spectrometer is used only for identification or for both – identification and quantification of SGs”.
We’re sorry for the mistake. This information is now added in the method description.
- “The chromatogram in Figure 1 does not give any information since the peaks of SGs and the integration baseline are not marked”.
The peak labels are missing in this figure. We now inserted the correct version of the chromatogram with peak labels. We’re sorry for the mistake. The integration marks of all peaks cannot be shown at the same time by the MassLynx software. This software allows to see only one integrated peak in a chromatogram. Therefore, we prefereed to exclude the integration marks instead of showing just one integrated peak.
- “Reference for Figure 1 (line 234) states differences in SGs concentrations however Figure 1 has nothing to do with this information”.
We’re sorry for the mistake. We moved the reference to Figure 1 from line 234 (now 246) to the Materials and Methods section (2.2).
- “The given result units (GAE) are different in result chapter (GAE/g DM)”.
Corrected
- “In the Results and Discussion, unrelated information is given in the last paragraphs of chapters 3.1.2-3, 3.1.6. Radical ABTS abbreviation is given in three different forms in lines 18, 122, 124 . Also, two terms are used for the same parameter – radical scavenging and radical inhibition (lines 286,289). The terms „fat“ and „lipids“ are used as equivalents (lines 169, 170)”.
Corrected in the text
Now line 18, 127, 129 - ABTS●+ free radical ….; 306, 308, 309 - inhibition scavenging;
177, 178 - Crude fat Lipids.
- “SGs rebaudiosides B and F, steviolbioside were not detected by authors, however they are stated as „remaining“, i.e. present after dominant stevioside and rebaudioside” A.”
The rebaudiosides B and F, and steviolbioside could be detected by mass spectrometry, but their UV peaks were too small for quantitation or interfered by matrix signals. Therefore, in table 2 the value 0 is given. Tis is now changed to „n.q.“ = not quantifiable. The sentence about the „remaining steviolglycosides“ in rows 243 (now 256) is about the results in the cited papers in row 241 (now 253).
Reviewer 4 Report
Manuscript Number: applsci- 1158894
Title: Comparative assessment of the basic chemical composition and
antioxidant activity of Stevia rebaudiana Bertoni dried leaves, grown in
Poland, Paraguay and Brazil
Overview and general recommendation
The title is specific and reflects the main ideas of the article. The article structure is compact, sequential and logical.
In my opinion in the Materials and methods section there is insufficient information on the samples of Stevia leaves studied (their origin, drying method, the company that sells it etc.).
In terms of Results and Discussion, a graphical representation of the values regarding the antioxidant activity would have better highlighted the differences between the samples. It is not sufficiently explained why these differences were obtained. No explanation is given for the data in Figure 1.
The conclusions are insufficient for the results obtained without arguments on the data obtained. Where will these results be used in the future?
Major comments:
- I consider that the part of Materials and Methods the must be reviewed by adding information on sampling.
- I consider that the part of Results and discussion the must be reviewed, by adding a figure on antioxidant activity and to improve explanations of differences in data obtained.
- I recommend that the part of Conclusions, the must be improved them by adding arguments about the results obtained.

Author Response
Reviewer 4
Authors are grateful for the Reviewer comments. The answer to the comments are below.
- “The title is specific and reflects the main ideas of the article. The article structure is compact, sequential and logical”.
- “In my opinion in the Materials and methods section there is insufficient information on the samples of Stevia leaves studied (their origin, drying method, the company that sells it etc.)”.
- We had researched dried stevia leaves, as a ready to use commercial product, that were purchased from Poland in direct trade points via the Internet. Polish dried stevia directly was bought from the producer, but dried Paraguay and Brazilian stevia through an online store. The seller were able give the country of origin of stevia but not the part of region were the stevia was agricultured. Shops run commercial activities, therefore mentioning the company names would be inappropriate, because of free advertising for them.
Line 89-91. Some information has been added in manuscript body - Polish dried leaves were bought directly from the producer via the Internet, but dried Paraguayan and Brazilian Stevia through an online store.
- “In terms of Results and Discussion, a graphical representation of the values regarding the antioxidant activity would have better highlighted the differences between the samples. It is not sufficiently explained why these differences were obtained”.
We are grateful for this comment albeit in figures is pretty often difficult to read correctly the value of results and they take a lot of space. Instead of 1 table we will have 3 figures that takes much more space. In our opinion values in the table give as much more possibility to make conclusions.
Line 297-302 The following explanation has been added - Differences between high antioxidant activity in Paraguayan and Brazilian Stevia and, at the same time, lover content of polyphenolic compounds, vitamin C, compared to Polish Stevia, can be explain by higher content of other antioxidants that were not measured, for example fat soluble (sesquiterpenes - δ-caryophyllene, δ-farnesene, humulene, candinene, nerolidol and monoterpenes - linalool, terpinen-4-ol, terpineol), chlorofil, xanfofils and others.
- “No explanation is given for the data in Figure 1.”
Corrected
Explanation for the data in Figure 1 is given.
- “The conclusions are insufficient for the results obtained without arguments on the data obtained. Where will these results be used in the future?”
Line 353-354 The following explanation has been added - Results of this research could be used in future animal or human studies to evaluate, for example, antioxidant and anti-inflammatory properties.
Round 2
Reviewer 2 Report
The reviewer appreciates the fast reply and detailed explanation for the questions in the previous review process. The comments for the answers to those 12 questions are as below:
1 & 2. The question was not regarding the commercial activities or sellers’ credential. This paper intended to compare the dried leaves of Stevia rebaudiana Bertoni cultivated from three different countries, so the most important thing is to confirm whether the samples were really come from these three countries. First, “purchased from online store in that country” does not equal to “collected the samples from that country”. Secondly, without the detailed information about the samples collected, this research implied that all the samples from a certain country can be seen as the same, no matter the seasons, the soil conditions, the processing methods. However, the variation caused by the above factors are higher than the cultivated countries of samples. Tracing the source is important in most of disciplines.
- Duplicate and triplicate means the technical replicates, right? How about the biological replicates, i.e., how many samples were purchased from each country?
- The reviewer meant how the paper can help or inspire the readers for their research. For example, which countries’ samples were suggested to be used for better performance, or what might be the factor to cause the differences among the different countries. For the claims authors added:” Results of this research could be used in the future animal or human studies to evaluate, for example, antioxidant and anti-inflammatory properties”, many studies have been done before this paper.
- Did the author use the chromatogram in Figure 1 for quantification? How reliable are the results do authors think about the compounds such as Rebaudioside D? If only [M-H]- SIM mode was used, how did the author distinguish the pair of Rebaudioside A and Rebaudioside E, or the pair of Rebaudioside B and Stevioside? They share the same molecular weight.
- Reviewer did not find the sentences that mentioned the way of calculation the digestible carbohydrates in the reference [39]. What the reviewer asked was to present the related sentences shown in the reference [39], not to rewrite the sentence in this manuscript.
- This question has been addressed appropriately.
- Similar as question 6. Reviewer checked the original references and did not find some data mentioned in this paper. Please provide the exact sentence in these references that mention the reported value, e.g., 94.6g/100g. Otherwise, the values and data listed here would be considered as fraud.
9-11. This manuscript is not a review paper, so it is not necessary to mention all the other components that the authors did not test. If those components are important, then the authors should repeat the similar experiments to detect their concentration in this study.
- Is this mistake just a typo? Or the analysis method had some flaws?
For the last comments by authors, reviewer suggests that they should combine the results from this study and the results from the animal experiment, and then resubmit to exhibit a better quality and significance.
Author Response
Authors are grateful for this all comments. Thank you very much.
The answers as below.
1 & 2. The question was not regarding the commercial activities or sellers’ credential. This paper intended to compare the dried leaves of Stevia rebaudiana Bertoni cultivated from three different countries, so the most important thing is to confirm whether the samples were really come from these three countries. First, “purchased from online store in that country” does not equal to “collected the samples from that country”. Secondly, without the detailed information about the samples collected, this research implied that all the samples from a certain country can be seen as the same, no matter the seasons, the soil conditions, the processing methods. However, the variation caused by the above factors are higher than the cultivated countries of samples. Tracing the source is important in most of disciplines.
The origin was confirmed by the bar codes of the product on the labels of the original packaging.
Results of these studies can be an introduction and inspiration to the other researchers. Observed differences in composition, in particular regarding the content of steviol glycosides and antioxidant activity, could be confirmed (or unconfirmed) by further study, assessing simultaneously (in one paper) composition of this product from several countries, depending on the cultivation country-specific conditions.
- Duplicate and triplicate means the technical replicates, right? How about the biological replicates, i.e., how many samples were purchased from each country?
A 5 kg consumer package product was purchased from each country and samples were taken from it.
- The reviewer meant how the paper can help or inspire the readers for their research. For example, which countries’ samples were suggested to be used for better performance, or what might be the factor to cause the differences among the different countries. For the claims authors added:” Results of this research could be used in the future animal or human studies to evaluate, for example, antioxidant and anti-inflammatory properties”, many studies have been done before this paper.
Previously, the composition of a commercial stevia product from different countries was never simultaneously compared in one study.
At this stage of this experiments, it is impossible to say which countries’ samples could be suggested to used for better performance.
Authors of these studies conducted an experiment on experimental animals and used stevia (from three different, taken into account countries) as addition to the rat's diet. Results of these studies are currently being analyzed and prepared for publication.
As mentioned above, data obtained in this study may inspire further researchers to assess composition of this product depending on the cultivation country-specific conditions, drying methods and storage condition.
Therefore, the last sentence of the chapter Results and Discussion is given:
Line 324-328 „As it is shown in this study and in the cited publications, the values of the composition of S. rebaudiana are within a wide range. Cultivation environmental factors such as temperature, sunlight exposure, soil composition and pH, fertilization, planting density, growth phase and plant physiology strongly affect chemical composition. Additionally drying method and storage conditions can modify bioactive compounds content [5,64].
- Did the author use the chromatogram in Figure 1 for quantification? How reliable are the results do authors think about the compounds such as Rebaudioside D? If only [M-H]- SIM mode was used, how did the author distinguish the pair of Rebaudioside A and Rebaudioside E, or the pair of Rebaudioside B and Stevioside? They share the same molecular weight.
It is of course true that UV detection at 200 nm is not very specific. Anyhow, MS shows, where to find the right UV peak. No one can absolutely exclude that there is matrix with the same UV absorption at the same retention time. Anyhow, if there is such an interfering matrix peak, it can be assumed to be similar in all samples and therefore at least the relative quantitation and comparing of the samples is possible.
The isobaric pairs Rebaudiosides A/E and Rebaudioside B/Stevioside can be distinguished by their different retention time. This is now clarified in the materials and methods section. Details can be found in my paper "Tandem mass spectrometric fragmentation patterns of known and new steviol glycosides with structure proposals" Rapid Commun. Mass Spectrom. 2011, 25, 1575– 1582, DOI: 10.1002/rcm.5024
It was
The mass spectrometer was used for confirmation of peak identity.
Now is
The peaks werde identified by retention time and by mass spectrometry in SIM mode by m/z of the [M–H]– ions.
- Reviewer did not find the sentences that mentioned the way of calculation the digestible carbohydrates in the reference [39]. What the reviewer asked was to present the related sentences shown in the reference [39], not to rewrite the sentence in this manuscript.
Sorry for this misunderstanding
It was
Digestible carbohydrates content was calculated by subtracting the sum of protein, crude fat, ash and dietary fiber from the dry matter, which was take as the 100 % [38 39].
Now line 100-102
Now is
Total carbohydrate content was calculated by subtracting the sum of total fat, protein, moisture and ash from the wet matter, which was take as the 100 %. Digestible carbohydrates were calculated by subtracting dietary fiber from total carbohydrate [39].
- This question has been addressed appropriately.
No comments
- Similar as question 6. Reviewer checked the original references and did not find some data mentioned in this paper. Please provide the exact sentence in these references that mention the reported value, e.g., 94.6g/100g. Otherwise, the values and data listed here would be considered as fraud.
Line 147 / Now line 149
It was
The values reported in the literature were lower: 95.4 g/100 g [36, 45]; 94.6 g/100 g [6, 9]……….
Now is
The values reported in the literature were lower: 95.4 g/100 g [36, 45]; 94.6 g/100 g [6]……………..
In the cited literature [6], the moisture of dried stevia leaves was given = 5,37 g/100 g
9-11. This manuscript is not a review paper, so it is not necessary to mention all the other components that the authors did not test. If those components are important, then the authors should repeat the similar experiments to detect their concentration in this study.
Authors treated this additional information as an important supplement to the obtained data in this study. If it is possible, please give us your agreement to remain them.
- Is this mistake just a typo? Or the analysis method had some flaws?
We are very sorry, it was only the technical mistake
For the last comments by authors, reviewer suggests that they should combine the results from this study and the results from the animal experiment, and then resubmit to exhibit a better quality and significance.
This is a very good idea. But the problem is that we have to put too many results from animal studies into one publication. They don't fit.
Authors are grateful for this all comments. Thank you very much.
The answers as below.
1 & 2. The question was not regarding the commercial activities or sellers’ credential. This paper intended to compare the dried leaves of Stevia rebaudiana Bertoni cultivated from three different countries, so the most important thing is to confirm whether the samples were really come from these three countries. First, “purchased from online store in that country” does not equal to “collected the samples from that country”. Secondly, without the detailed information about the samples collected, this research implied that all the samples from a certain country can be seen as the same, no matter the seasons, the soil conditions, the processing methods. However, the variation caused by the above factors are higher than the cultivated countries of samples. Tracing the source is important in most of disciplines.
The origin was confirmed by the bar codes of the product on the labels of the original packaging.
Results of these studies can be an introduction and inspiration to the other researchers. Observed differences in composition, in particular regarding the content of steviol glycosides and antioxidant activity, could be confirmed (or unconfirmed) by further study, assessing simultaneously (in one paper) composition of this product from several countries, depending on the cultivation country-specific conditions.
- Duplicate and triplicate means the technical replicates, right? How about the biological replicates, i.e., how many samples were purchased from each country?
A 5 kg consumer package product was purchased from each country and samples were taken from it.
- The reviewer meant how the paper can help or inspire the readers for their research. For example, which countries’ samples were suggested to be used for better performance, or what might be the factor to cause the differences among the different countries. For the claims authors added:” Results of this research could be used in the future animal or human studies to evaluate, for example, antioxidant and anti-inflammatory properties”, many studies have been done before this paper.
Previously, the composition of a commercial stevia product from different countries was never simultaneously compared in one study.
At this stage of this experiments, it is impossible to say which countries’ samples could be suggested to used for better performance.
Authors of these studies conducted an experiment on experimental animals and used stevia (from three different, taken into account countries) as addition to the rat's diet. Results of these studies are currently being analyzed and prepared for publication.
As mentioned above, data obtained in this study may inspire further researchers to assess composition of this product depending on the cultivation country-specific conditions, drying methods and storage condition.
Therefore, the last sentence of the chapter Results and Discussion is given:
Line 324-328 „As it is shown in this study and in the cited publications, the values of the composition of S. rebaudiana are within a wide range. Cultivation environmental factors such as temperature, sunlight exposure, soil composition and pH, fertilization, planting density, growth phase and plant physiology strongly affect chemical composition. Additionally drying method and storage conditions can modify bioactive compounds content [5,64].
- Did the author use the chromatogram in Figure 1 for quantification? How reliable are the results do authors think about the compounds such as Rebaudioside D? If only [M-H]- SIM mode was used, how did the author distinguish the pair of Rebaudioside A and Rebaudioside E, or the pair of Rebaudioside B and Stevioside? They share the same molecular weight.
It is of course true that UV detection at 200 nm is not very specific. Anyhow, MS shows, where to find the right UV peak. No one can absolutely exclude that there is matrix with the same UV absorption at the same retention time. Anyhow, if there is such an interfering matrix peak, it can be assumed to be similar in all samples and therefore at least the relative quantitation and comparing of the samples is possible.
The isobaric pairs Rebaudiosides A/E and Rebaudioside B/Stevioside can be distinguished by their different retention time. This is now clarified in the materials and methods section. Details can be found in my paper "Tandem mass spectrometric fragmentation patterns of known and new steviol glycosides with structure proposals" Rapid Commun. Mass Spectrom. 2011, 25, 1575– 1582, DOI: 10.1002/rcm.5024
It was
The mass spectrometer was used for confirmation of peak identity.
Now is
The peaks werde identified by retention time and by mass spectrometry in SIM mode by m/z of the [M–H]– ions.
- Reviewer did not find the sentences that mentioned the way of calculation the digestible carbohydrates in the reference [39]. What the reviewer asked was to present the related sentences shown in the reference [39], not to rewrite the sentence in this manuscript.
Sorry for this misunderstanding
It was
Digestible carbohydrates content was calculated by subtracting the sum of protein, crude fat, ash and dietary fiber from the dry matter, which was take as the 100 % [38 39].
Now line 100-102
Now is
Total carbohydrate content was calculated by subtracting the sum of total fat, protein, moisture and ash from the wet matter, which was take as the 100 %. Digestible carbohydrates were calculated by subtracting dietary fiber from total carbohydrate [39].
- This question has been addressed appropriately.
No comments
- Similar as question 6. Reviewer checked the original references and did not find some data mentioned in this paper. Please provide the exact sentence in these references that mention the reported value, e.g., 94.6g/100g. Otherwise, the values and data listed here would be considered as fraud.
Line 147 / Now line 149
It was
The values reported in the literature were lower: 95.4 g/100 g [36, 45]; 94.6 g/100 g [6, 9]……….
Now is
The values reported in the literature were lower: 95.4 g/100 g [36, 45]; 94.6 g/100 g [6]……………..
In the cited literature [6], the moisture of dried stevia leaves was given = 5,37 g/100 g
9-11. This manuscript is not a review paper, so it is not necessary to mention all the other components that the authors did not test. If those components are important, then the authors should repeat the similar experiments to detect their concentration in this study.
Authors treated this additional information as an important supplement to the obtained data in this study. If it is possible, please give us your agreement to remain them.
- Is this mistake just a typo? Or the analysis method had some flaws?
We are very sorry, it was only the technical mistake
For the last comments by authors, reviewer suggests that they should combine the results from this study and the results from the animal experiment, and then resubmit to exhibit a better quality and significance.
This is a very good idea. But the problem is that we have to put too many results from animal studies into one publication. They don't fit.
Reviewer 3 Report
The manuscript parts of methodology and results have been improved; however, the novelty and the aim of the study are not explained or highlighted and are kept unchanged by authors. A single sample from each country is not enough to make a comparison among different countries especially when cultivation conditions are unknown. The authors could substantiate the aim from the view of customers or further dry leaf material application, etc. The discussion and conclusions should be adapted to the aim then.
The term “essential oil” is changed to “lipid fraction” in lines 54-55 which is incorrect.
In the Methods, vitamin C determination is left for some reason in the „Proximate composition“ chapter.
In the Results and Discussion, unrelated information is left in the last paragraphs of chapters 3.1.2-3. In 3.1.6. (line 234-236), the connection of the sentence (“Based on the literature data [11] S. rebaudiana contain a high level of oxalic acid (2295 mg/100 g DM), which reduces the bioavailability of calcium, iron and other minerals.”) to the preceding text is unclear. What bioavailability is meant - to stevia or to human consuming stevia?
Unnecessary capitalization of the word "stevia" is used.
Author Response
Authors are grateful for this all comments. Thank you very much.
The answers as below.
- The manuscript parts of methodology and results have been improved; however, the novelty and the aim of the study are not explained or highlighted and are kept unchanged by authors.
Novelity - Previously, the composition of a commercial stevia product from different countries was not simultaneously compared in one study.
Aim - Line 353-354 We added following sentence „Results of this research could be used in future animal or human studies to evaluate, for example, antioxidant and anti-inflammatory properties”.
Authors of these studies have just conducted experimental animal studies using stevia from three different countries considered as a supplement to rats' diets. The results of these studies are currently being analyzed and prepared for publication.
Otherwise, results of these studies can be an introduction and inspiration to the other researchers. Observed differences in composition, in particular regarding the content of steviol glycosides and antioxidant activity, could be confirmed (or unconfirmed) by further study, simultaneously assessing composition of this product from several countries, depending on the cultivation country-specific conditions.
Line 324-328 - Therefore, the last sentence of the chapter Results and Discussion is given:
„As it is shown in this study and in the cited publications, the values of the composition of S. rebaudiana are within a wide range. Cultivation environmental factors such as temperature, sunlight exposure, soil composition and pH, fertilization, planting density, growth phase and plant physiology strongly affect chemical composition. Additionally drying method and storage conditions can modify bioactive compounds content [5,64]”.
- A single sample from each country is not enough to make a comparison among different countries especially when cultivation conditions are unknown.
A 5 kg consumer package product was purchased from each country and samples were taken from it.
As mentioned above, results obtained in this studies can be an introduction to the other one, simultaneously (in one paper) assessing composition of this product from different countries, depending on the cultivation country-specific conditions, drying methods and storage condition.
- The authors could substantiate the aim from the view of customers or further dry leaf material application, etc. The discussion and conclusions should be adapted to the aim then.
As we mentioned above, previously the composition of commercial stevia product from different countries was not simultaneously compared in the one study.
In discussion , line 353-354, we added the following sentence „Results of this research could be used in future animal or human studies to evaluate, for example, antioxidant and anti-inflammatory properties”.
- The term “essential oil” is changed to “lipid fraction” in lines 54-55 which is incorrect.
Termin „olejek eteryczny” zmieniono na „frakcja lipidowa” w wierszach 54–55, co jest niepoprawne.
The introduction includes, i.a. data on the composition of stevia, both the studied and the non-tested, in order to perform a short, but at the same time more extensive characterization of this plant. The authors ask for the possibility not to delete this sentence, if possible.
Sorry for this misunderstanding
It was Essential oils Lipid fraction of Stevia, being contains sesquiterpenes……
The authors included the previous correct sentence – line 54
Now is Essential oils of Stevia, being sesquiterpenes……
- In the Methods, vitamin C determination is left for some reason in the „Proximate composition“ chapter.
Title is changed
It was
2.1. Proximate composition
Now is - line 87
2.1. Proximate composition and vitamin C content
6. In the Results and Discussion, unrelated information is left in the last paragraphs of chapters 3.1.2-3.
Authors treated this information as an important supplement to the data obtained in this study. If it is possible, please give us your agreement to remain them.
- In 3.1.6. (line 234-236), the connection of the sentence (“Based on the literature data [11] rebaudiana contain a high level of oxalic acid (2295 mg/100 g DM), which reduces the bioavailability of calcium, iron and other minerals.”) to the preceding text is unclear. What bioavailability is meant - to stevia or to human consuming stevia?
Bioavailability concerns the human body
It was
Based on the literature data [11] S. rebaudiana contain high level of oxalic acid (2295 mg/100 g DM), which reduces bioavailability of calcium, iron and other minerals.
Now is - line 237
Based on the literature data [11] S. rebaudiana contain high level of oxalic acid (2295 mg/100 g DM), which reduces human bioavailability of calcium, iron and other minerals.
- Unnecessary capitalization of the word "stevia" is used.
We are very sorry, but in the other publications stevia is also written with a capital letter, e.g.
Lemus-Mondaca, R.; Vega-Gálvez, A.; Zura-Bravo, L.; Ah-Hen, K. Stevia rebaudiana Bertoni, source of a high-potency 384 natural sweetener: A comprehensive review on the biochemical, nutritional and functional aspects. Food Chem. 2012, 385 132, 1121–1132. doi:10.1016/j.foodchem.2011.11.140
Abou-Arab, A.E.; Abou-Arab, A.A.; Abu-Salem, M.F. Physico-chemical assessment of natural sweeteners steviosides 375 produced from Stevia rebaudiana Bertoni plant. Afr. J. Food Sci. 2010, 4, 269–281. http://www.academicjournals.org/ajfs.
Reviewer 4 Report
The authors have made all the requested corrections and the article can be published in its current form.
Author Response
The authors have made all the requested corrections and the article can be published in its current form.
Authors are grateful for this comments. Thank you very much.